# Morphophysiological Characterisation of Guayule (*Parthenium argentatum* A. Gray) in Response to Increasing NaCl Concentrations: Phytomanagement and Phytodesalinisation in Arid and Semiarid Areas

**DOI:** 10.3390/plants13030378

**Published:** 2024-01-27

**Authors:** Daniela Di Baccio, Aurora Lorenzi, Andrea Scartazza, Irene Rosellini, Elisabetta Franchi, Meri Barbafieri

**Affiliations:** 1Research Institute on Terrestrial Ecosystems, National Research Council of Italy (IRET-CNR), Via G. Moruzzi 1, 56124 Pisa, Italy; lorenzi.aurora0@gmail.com (A.L.); andrea.scartazza@cnr.it (A.S.); irene.rosellini@cnr.it (I.R.); 2National Biodiversity Future Center (NBFC), Piazza Marina 61, 90133 Palermo, Italy; 3Eni S.p.A., R&D Environmental & Biological Laboratories, Via Maritano 26, San Donato Milanese, 20097 Milan, Italy; elisabetta.franchi@eni.com

**Keywords:** saline and hypersaline conditions, growth analysis, biomass partitioning, photosynthetic pigments, chlorophyll fluorescence, mineral composition, sodium chloride, sodium translocation, sodium concentration factor, phytodesalinisation

## Abstract

Water and soil salinity continuously rises due to climate change and irrigation with reused waters. Guayule (*Parthenium argentatum* A. Gray) is a desert perennial shrub native to northern Mexico and the southwestern United States; it is known worldwide for rubber production and is suitable for cultivation in arid and semiarid regions, such as the Mediterranean. In the present study, we investigated the effects of high and increasing concentrations of sodium chloride (NaCl) on the growth and the morphophysiological and biochemical characteristics of guayule to evaluate its tolerance to salt stress and suitability in phytomanagement and, eventually, the phytodesalinisation of salt-affected areas. Guayule originates from desert areas, but has not been found in salt-affected soils; thus, here, we tested the potential tolerance to salinity of this species, identifying the toxicity threshold and its possible sodium (Na) accumulation capacity. In a hydroponic floating root system, guayule seedlings were subjected to salinity-tolerance tests using increasing NaCl concentrations (from 2.5 to 40 g L^−1^ and from 43 to 684 mM). The first impairments in leaf morphophysiological traits appeared after adding 15 g L^−1^ (257 mM) NaCl, but the plants survived up to the hypersaline conditions of 35–40 g L^−1^ NaCl (about 600 mM). The distribution of major cell cations modulated the high Na content in the leaves, stems and roots; Na bioconcentration and translocation factors were close to one and greater than one, respectively. This is the first study on the morphophysiological and (bio)chemical response of guayule to different high and increasing levels of NaCl, showing the parameters and indices useful for identifying its salt tolerance threshold, adaptative mechanisms and reclamation potential in high-saline environments.

## 1. Introduction

Water and soil salinity is one of the most relevant environmental factors which limit the productivity of plants and threaten life in ecosystems. The global effects of salinity are aggravated by the scarcity of water resources, soil consumption, climate change and the increasing human population worldwide [1,2]. The increase and accumulation of salt in waters and soils can occur under all climatic conditions. However, it is more frequent in arid and semiarid regions, depending on the balance between the water supply and evapotranspiration rate. This phenomenon is known as salinisation (and sodification if the main salt present is sodium chloride, or NaCl), and can be of natural and anthropogenic origin [3]. Natural causes of salinity include the weathering of rocks, the deposition of salts contained in rainfalls from evaporated oceans, seawater intrusion and capillary rise in coastal areas and low-lying countries. Human causes of salinity are directly related to agricultural practices, such as the overuse of fertilisers and irrigation with saline–sodic waters or salt-rich treated wastewaters, and the pumping of groundwater from wells with the raising of salts stored in the depth to the surface. Furthermore, the global rise in temperature and aquifer levels due to climate change accelerates desertification and land degradation, increasing the need to recover low-quality water for irrigation and to apply soil-restoration measures [4,5,6]. Soils are classified as saline when the ECe is 4 dS m^−1^ or more [7], equivalent to approximately 40 mM or 2.3 g L^−1^ NaCl. In this scenario, the continuous increase in salinisation (and sodification) transforms saline areas into hypersaline ones, which are extreme environments where the salinity is very high (similar or higher than that of seawater, 35–40 g L^−1^ NaCl or 600–680 mM), where plant growth is strongly limited and may be suitable for halophytes only [8,9,10].

Global soil mapping indicates that more than 900 million hectares are salt-affected, and salinisation is expected to increase continuously; with this trend, up to 50% of arable land could be drought- and salt-affected by 2050 [2,11]. In Europe, it is estimated that salt-affected areas comprise 24 Mha, approximately 2.05% of the total salt-affected soils worldwide [12], and these areas are mainly located in the Mediterranean region, along with Eastern Europe. Here, in the last two decades, seawater intrusions into lowlands near the coast, the growing frequency and magnitude of droughts and the improper management of irrigation water and drainage have increased the salinisation and degradation of soil and water resources [13].

The prevention and mitigation of soil and water salinity can be reached by reducing and/or removing the main causes of salt accumulation and adopting sustainable management practices in agroecosystems. One of these strategies is selecting plants/crops able to counteract salt stress by surviving and thriving in high-saline conditions. The simultaneous reclamation/revitalisation of salt-affected areas with plants able to absorb excessive salts and improve the soil structure (i.e., phytodesalinisation and phytomanagement) could be a cost-effective and environmental benefit contributing to the reduction in global salinity concerns [14,15,16].

Guayule (*Parthenium argentatum* A. Gray) is a perennial shrub from the Asteraceae family indigenous to the Chihuahuan desert in Mexico and the southern Texas; since the beginning of the twentieth century, this plant species has received attention for its capacity to biosynthesise and store rubber in the stems [17]. This commercial interest, together with an early use of guayule for fuel in Mexican smelters, progressively led to the depletion of the native populations and the consequent need to domesticate the plant [18]. Guayule cultivation was first developed in California and Arizona, and then extended to the other side of the Atlantic Ocean in Europe. Here, this shrub is well suited to the arid and semiarid regions of Mediterranean areas, as it grows within a range of temperatures from −15 to 40 °C and has an annual rainfall requirement of 250 to 380 mm [19,20]. From the end of the 1930s, the interest in guayule also bloomed in Italy, where the cultivation and research on the production of natural rubber continued, with ups and downs through World War II, the economic boom of the 1950s–1970s and the market competition with the *Hevea* rubber from Asia [20,21]. More recently, some circumstances have renewed the interest in the cultivation of guayule. One is the discovery that guayule is a source of natural hypoallergenic latex because it is characterised by a low protein content and, therefore, can replace the allergy-causing *Hevea* latex [21]. The second is the launch of some US and EU initiatives and projects on the economic potential of renewable resources from nonfood or industrial crops, and the production and exploitation of alternative rubber and latex sources (e.g., EPOBIO, 2005–2007; EU-PEARLS, 2008–2012). In this respect, the first tyres made entirely of natural rubber from guayule were produced by Bridgestone in 2015, and Cooper Tires and Panaridus developed plantations in Arizona with the same purpose [22]. Moreover, guayule, as a perennial desert nonfood shrub, can be grown on marginal lands with scarce water, any pesticides and low costs. In addition to natural rubber and latex, a wide range of co- and byproducts (including resins, pharmaceuticals, cosmetics and perfume) from guayule are under investigation and can be valorised, together with the use of plant residues and bagasse from rubber latex for biofuels [23,24,25].

Guayule, being native to the Chihuahuan desert, spontaneously grows in arid ecosystems, and so has a history of natural exposure to several environmental stresses (drought, low night temperatures, insect pests), but it is not found in salt-affected soils, so this species has no natural selection for salinity tolerance [17,26,27]. With the domestication and cultivation of guayule, research has revealed that, although this shrub is drought-tolerant, it must be irrigated for maximum sustained production [18,28,29]. Some studies have shown that guayule is susceptible to salt at germination, emergence and the development of seedlings [30,31]; this condition can be overcome by transplanting [18]. However, irrigation is a critical factor influencing the establishment and rubber production of guayule in the current arid and semiarid regions of cultivation; here, the supply of waters containing soluble salts can aggravate the increasing salinity of the soils. Guayule can be grown under saline irrigation conditions [27,32,33], but its potential to tolerate high salt concentrations, its adaptation mechanisms and its salt tolerance threshold have not been specifically investigated yet.

In this work, we studied the effects of high and increasing NaCl concentrations on the morphophysiological and biochemical characteristics of *P*. *argentatum*. With this purpose, a screening procedure in a hydroponic floating root system was developed. Here, NaCl concentrations from 2.5 to 40 g L^−1^ were tested to simulate the global salinity increase towards hypersaline conditions without the indirect effects of an impaired soil structure. In previous experiments [27,33], the highest salt treatments tested on guayule consisted of saline irrigation waters with an EC not higher than 10 dS m^−1^ (~5 g L^−1^ NaCl). In any case, such salt levels were not completely available for the plant, as in a hydroponic growth system. Thus, this is the first screening on the response of guayule to high and increasing concentrations of NaCl, reaching levels close to that of seawater. Moreover, few non-halophytic industrial crops comparable to guayule, such as Jerusalem artichoke or cotton [34,35], have been subjected to elevated saline conditions (>7 g kg^−1^ soil and up to 12 g L^−1^, respectively).

The guayule tolerance to NaCl stress was assessed by monitoring the growth and photosynthetic parameters during a one-month screening at the vegetative stage. Destructive and chemical determinations at the beginning and end of the test supported the models of the growth analysis and leaf chlorophyll variations, and verified the distribution of the main nutrients and sodium (Na). The specific goals of this research were: (i) to shed light on the response mechanism of guayule to salt stress; (ii) to identify the NaCl salinity threshold at which this species can survive and thrive; (iii) to investigate the absorption, distribution and accumulation capacity of Na from phytomanagement and/or phytodesalinisation perspectives.

## 2. Results

### 2.1. Nondestructive Determinations: Monitoring of the Morphological, Chlorophyll Fluorescence and Gas-Exchange Parameters during the Test

During the NaCl tolerance test (Figure 1), no variation was observed for the root maximum length between the control and the treated guayule plants at any NaCl treatment tested (Figure 2A). Indeed, the root length of the controlled plants increased from 8.3 to 11.0 cm, while, in the treated plants, the root growth was inhibited and remained stable (7.5–8.3 cm) throughout the test. The number of leaves started to reduce (−22%) between 10 and 15 g L^−1^ NaCl (i.e., between 13 and 15 days of the NaCl treatment) until the end of the experiment (30 days of growth and 40 g L^−1^ NaCl), reaching a decrease of more than 60% (Figure 2B).

The mean leaf area decreased between 20 and 25 g L^−1^ NaCl (about −30%), progressively reducing from 5.7 to 2.1 cm^2^ (Figure 3A). Nondestructive measurements of the leaf area during the NaCl tolerance test were thanks to the construction of a regression model between the leaf morphological traits (orthogonal diameters: length × width, L × W) and the scanning of single leaves at the beginning and the end of the experiments (Appendix A). A visual diagram illustrating the differences in growth between the control and the NaCl-treated plants is displayed in Appendix A.

The leaf total chlorophyll (Chl *a* + *b*) dropped at the NaCl dose of 2.5 g L^−1^ and went on decreasing until 15 g L^−1^ NaCl (i.e., 15 days of treatment), when it reached the maximum reduction (−86%) shown during the NaCl treatments (Figure 3B). After that, there was a gradual recovery of the chlorophyll content, which reached 15.72 µg cm^−2^. However, such a level was almost half of that reached by the control (Figure 3B).

Nondestructive measurements of Chl *a* + *b* during the NaCl tolerance test were thanks to the construction of a regression model between the SPAD values and the chemical extractions of the chlorophylls performed at the beginning and the end of the experiments. The leaf Chl *a* + *b* content varied from 18.1 to 67.0 SPAD relative unit (r. u.) in the controls and from 30.0 to 75.3 SPAD r. u. in NaCl-treated plants (Appendix A).

The leaves of the NaCl-treated guayule maintained the same Fv/Fm level as the control until 15 g L^−1^ NaCl, when it tended to decrease, becoming significantly different (*p* < 0.05) in the presence of 25 g L^−1^ NaCl. Then, the Fv/Fm level went on decreasing by 69.5 and 87% at 30 and 40 g L^−1^ NaCl, respectively (Figure 3C). The Φ_PSII_ started to reduce from 15 to 25 g L^−1^ NaCl (an average decrease of −19%), and then dropped in the presence of both 30 and 40 g L^−1^ NaCl treatments (Figure 3D).

The gas-exchange parameters (Figure 4) were measured in the presence of 15 g L^−1^ NaCl (i.e., 15 days of treatment), the concentration threshold at which the highest impairments of the leaf traits were observed and the efficiency of the PSII photochemistry started to decline. The net CO_2_ assimilation rate (A) decreased (about −63%), as well as the gs (−58%) and E (−59%) in the NaCl-treated guayule plants (Figure 4A–C). The intercellular CO_2_ concentration (Ci), A/E and A/gs ratios did not vary in the salt-stressed plants compared to the control (Figure 4D–F).

### 2.2. Destructive Determinations: Growth, Biomass Partitioning and Water Content

At the end of the NaCl treatment, the control and treated plants of guayule were harvested and separated into the leaves, stems and roots for the destructive determinations, such as the growth analysis and mineral composition. In the plants exposed to increasing NaCl concentrations up to 40 g L^−1^ NaCl, the DW of the leaves and stems decreased by 47.5 and 36.5%, respectively, with a total biomass reduction of 45.6% (Figure 5).

The FW/DW of the leaves was about three-fold lower than in the control, and the leaf RWC was affected by a 63% decrease (Table 1). The LMA was 2.6-fold higher in the NaCl-treated plants than in controls, whereas the LAR was 3.0-fold lower; the LMR and NAR did not change (Table 1). In the stem, only the FW to DW ratio was decreased (−20%) by the NaCl treatment, while no significant variation was observed in the roots (Table 1). In the whole plant of guayule, the LA was dramatically reduced (−82%) by the NaCl treatment, and both the FW/DW and RGR were halved. The shoot-to-root ratio, as well as the stem/leaf ratio, did not change (Table 1).

#### 2.2.1. Photosynthetic Pigments and Nutrients

The Chl *a*, Chl *b* and carotenoids were negatively affected by the treatment with increasing NaCl concentrations up to 40 g L^−1^ (Figure 6A); the Chl *a*/*b* was reduced by half compared to the control and the ratio of total chlorophyll (Chl *a* + *b*) to carotenoids was not affected (Figure 6B). The concentration of K, P, Ca and Mg at the end of the experiment was determined in the leaves, stems and roots of the control and NaCl-treated plants (Table 2). Potassium and Ca decreased in the roots (−68%) and stems (−51.5%) of guayule after NaCl exposure, whereas the P did not change in any organs; the Mg concentration in the leaves of the NaCl-treated plants enhanced (2.3-fold higher) compared to the control (Table 2).

#### 2.2.2. Sodium Concentration and Content: Distribution, Translocation, Accumulation and Relation with the Major Cations

After the exposure to increasing NaCl concentrations up to 40 g L^−1^, the Na concentration in the leaves, stem and roots of guayule was strongly enhanced, reaching about the same level (36,100–42,200 mg g^−1^ DW) in all the plant organs analysed (Figure 7A). The content of Na progressively increased from the root to the stem to the leaves of the treated guayule plants, with values drastically higher (1:6.6, 1:14 and 1:18.7, respectively) than the corresponding organs in the controls (Figure 7B). The Na translocation factor was more than 8- and 2.5-fold higher than in the control; the Na bioconcentration factor was 1.0 or close to 1.0 in all the organs analysed (Figure 7C).

The relative content of the major cations, K, Ca and Mg, to Na strongly decreased in the leaves, stems and roots of the NaCl-treated guayule compared to the control (Table 3). The K/Na, Ca/Na and Mg/Na ratios in the treated plants were relatively higher in the leaves than in stems and roots. In particular, the K/Na reached the same level in the stems and roots, where it was almost three-fold lower than in the leaves; both the Ca/Na and Mg/Na ratios followed the same trend: leaves > roots > stems (Table 3).

## 3. Discussion

In this work, we focused on the vegetative stage of first-year growth guayule plants to illustrate the mechanisms underlying their tolerance capacity in saline environments, from the morphophysiological to the (bio)chemical level. This phenological phase of active growth is fundamental for the plant establishment and the subsequent accumulation of rubber in the stem [18,36].

### 3.1. Morphophysiological Response to Increasing NaCl Concentrations

During the salt tolerance test, guayule plants were monitored daily for visual stress symptoms, and biometric and physiological parameters were performed on the shoots and roots every 2–5 days after each addition of NaCl.

The leaves started to change colour (from grey–green to greyer) after the 5 g L^−1^ NaCl treatment and, later (from 10 to 15 g L^−1^ NaCl), evident signs of early leaf senescence, such as turgor loss, edge browning and falling, were added to this colour change. Although inhibited in elongation compared to the control plants, the roots maintained their maximum length unchanged for the entire test duration (Figure 2A, Appendix A). The volume of the root apparatus was progressively reduced from 5 g L^−1^ NaCl, mostly due to the decreased number of lateral new roots, which also appeared thinner and less vigorous than the control when treated with the highest NaCl doses of 30 and 40 g L^−1^ (Appendix A). However, according to these first visual and general observations, in the guayule exposed to high and increasing NaCl concentrations, the leaves were the most impacted organs by the salinity. Several works show that the leaves and shoots are generally more sensitive to salinity than root growth, although dependent on the salt stress intensity, plant species and phenological stage [37].

Guayule plants of about 3.5 months old exposed to saline (NaCl > 40 mM or 2.34 g L^−1^) conditions and increasing NaCl concentrations (up to 40 g L^−1^) started to show reduced leaf numbers in the presence of 15 g L^−1^ NaCl and the expansion of the leaf blade after the addition of 25 g L^−1^ NaCl (Figure 2B and Figure 3A). The plant response to salinity stress is a two-phase process: a rapid, osmotic phase, which starts immediately after the roots perceive the increased external salt concentrations around them, and a slower ionic phase, in which Na enters the plant. The distinction between the effects of these two phases on plants is not so evident, and is clearly separated in time when the salinity reaches the high levels as employed in our case [37,38]. However, before the addition of 15 g L^−1^ NaCl, the guayule gradually experienced the osmotic component of the salt stress, with the inhibition of cell expansion in the roots and the reduction in its capacity to extract water and nutrients from the medium, but without a significant effect on the leaf growth rate. This means that no leaves had yet been damaged or fallen, or that any fallen leaves were replaced by younger developing leaves. Later, this turnover was disrupted by the growth inhibition in the young leaves and/or the induction of early senescence in the mature leaves. Over the following days, prematurely senescent leaves showed symptoms of injury and shedding, decreasing the total number of leaves observed in the treated plants. The inhibition of the development in young leaves was due to the reductions in cell elongation and division, leading to slower leaf appearance and a smaller final size (Figure 3A,B). These modifications resulted from the changes in the cell dimension (area and/or depth), leading to alterations in the form of the leaves, which often appeared shorter and thicker [39,40]. This is consistent with the drop in the mean area of the leaf blade recorded in guayule from 25 to 40 g L^−1^ NaCl (Figure 3A), accompanied by an increased LMA at 40 g L^−1^ (Table 1), which is commonly associated with a higher thickness of the leaf [41]. Moreover, the leaves of the guayule subjected to high NaCl concentrations (25–40 g L^−1^ or 428–684 mM NaCl) were smaller and thicker. However, they did not undergo significant morphological variations, as shown by the unchanged linear regression model between the product of the leaf orthogonal diameters (length × width, L × W) and the LA compared to the controls (Appendix A).

The leaf of guayule is generally ovate, entire and slightly acute, although its form can change according to the age, the amount of water available and position on the twig. Both the leaf adaxial and abaxial surfaces are covered with characteristic T-shaped trichomes, which give it the typical greyish-green appearance (hence, the epithet “*argentatum*”) and aid in reducing evapotranspiration [32,42]. Although guayule is a perennial plant species, the leaves are partially deciduous, since the lower part of them shrivel up and are shed during winter drought. This natural leaf behaviour may be a defence response mechanism against stress, since, under severe water shortages or lasting freezing temperatures, guayule enters a semidormant state characterised by darker, greyer leaves that can drop and then be replaced when resources become again available [32,36]. This could explain the changes in the colour and the shedding of older leaves observed in the guayule plants during our salt tolerance test.

The darker and thicker leaves of guayule under high salt concentrations (>15 g L^−1^ or 257 mM NaCl) made the measurements of the total leaf chlorophyll content with the SPAD more difficult. In fact, in this case, higher SPAD values were not always the result of higher total chlorophyll contents in the leaves; however, such difficulties were overcome by relating most of the SPAD values to the chemical extraction of Chl *a* and Chl *b*, and building up distinct regression models between the treated and control guayule plants (Appendix A).

The reduction in the leaf number at 15 g L^−1^ NaCl and in the leaf expansion at 25 g L^−1^ NaCl revealed that the growth of guayule started to be seriously impaired; thus, the rate at which new leaves were produced was slower than the rate at which the old leaves died [43,44]. Such an effect showed that the salt stress was affecting the plant’s primary metabolism, and its photosynthetic capacity was insufficient to supply the generation of new tissues. Indeed, the monitored leaf chlorophyll fluorescence parameters gradually started to decrease from 15 g L^−1^ NaCl, when the Chl *a* + *b* content also reached the lowest values (Figure 3B). At this NaCl level, the later and slower effects of salt stress on the plant biochemical and physiological responses were likely added to the immediate osmotic effects on the growth. To verify this hypothesis, we performed gas-exchange measurements, which are the most readily and nondestructive measurable parameters reflecting the response of the whole plant to salinity [37,45,46]. The high decrease in the stomatal conductance, and the consequent net CO_2_ assimilation and transpiration rate in the presence of 15 g L^−1^ NaCl, was consistent with the observed impact of salt on the photosynthetic efficiency (Figure 3C,D and Figure 4). Indeed, both Fv/Fm and Φ_PSII_ describe the state of the PSII, which determines the quantum efficiency of the photochemical transports and the heat dissipation capacity in photosynthesis; thus, the reduction in these parameters resulted in less energy for CO_2_ assimilation [47,48]. Moreover, despite a higher stomatal closure than the control at the same NaCl treatment, the intercellular CO_2_ concentration and A/E and A/gs ratios were not affected. This indicated that the decreased CO_2_ uptake was not counterbalanced by the efficient use of intercellular CO_2_ in the chemical reactions responsible for converting CO_2_ into glucose. The stomatal closure of the guayule leaves under 15 g L^−1^ NaCl can be considered as an osmotic-adjustment mechanism that limits leaf dehydration and maintains the instantaneous and intrinsic water-use efficiency. Still, it was insufficient to prevent the photoinhibition and/or photodamages of the PSII, with the consequent reduction in the photosynthetic activity. At such NaCl levels, with the onset of salinity stress, the ionic-specific phase of salt stress probably started to occur in addition to the osmotic phase, providing damage to several cell structures and biomolecules [44,49]. After the addition of 15 g L^−1^ NaCl, despite the decreasing efficiency of the PSII photochemistry, the Chl *a* + *b* content per leaf area apparently increased with the enhancement in the NaCl addition up to 40 g L^−1^ NaCl (Figure 3B). This paradox can be explained by the changes in the leaf morphology described above, which produced smaller and thicker leaves, resulting in a higher chloroplast density per unit of leaf area. Thus, when the total chlorophyll is expressed on a leaf unit area basis rather than the DW, an increase can also be measured in the salt-affected plants because of the highly decreased area and the increased density in the leaves due to the salinity. In contrast, the amount of chlorophyll per leaf or whole plant is always reduced. For these reasons, for the evaluation of the stress effects on the plants, the ratio of Chl *a*/Chl *b* or the leaf composition in photosynthetic pigments is considered a better indicator than the total chlorophyll content [46,50].

### 3.2. Biomass Partitioning, the Distribution of Nutrients and Sodium

The observed reduction in the guayule growth rate, leaf expansion, photosynthetic efficiency and chlorophyll content in response to high and increasing NaCl concentrations are general effects of plant responses to salt stress. As for other abiotic stress, it is difficult to understand whether the reduced rate of photosynthesis is the cause of the growth reduction or the result [37,40,51]. However, the plant biomass production and allocation to different organs, the growth analysis, the composition in photosynthetic pigments and the main minerals, including Na, can help to elucidate the mechanisms of guayule response to the high-saline conditions tested and its adaptation strategy (tissue tolerance and/or exclusion) for future perspectives in the phytomanagement of arid and semiarid regions.

At the end of the NaCl screening test, the strong decrease in the whole-plant DW of the treated guayule was due to the reduced biomass allocation in the leaves and stems, whereas the roots, although inhibited in growth, maintained a similar DW as the control (Figure 5). This confirmed that, under saline conditions, the growth of the root apparatus is usually less affected than the shoot [37,40]. As expected, the shoot of the NaCl-exposed plants was also dehydrated, with less extent in the stems than in the leaves, which lost their turgor, as shown by the decline of the RWC (Table 1). The reduced leaf area expansion resulted in a lower LAR, impacting the RGR of the entire plant [52]. However, the NAR, the other component of the RGR, was not affected, meaning that the DW gained per unit of leaf area every day remained unchanged despite the salt stress. Although having the typical characteristics of drought and winter-deciduous perennial shrubs, guayule has been classified as a C3 species [53]. The resistance to water stress of guayule originating from desert regions seems to be mostly related to some specific anatomical and morphological traits of the leaf [32]. The mesophyll consists of several layers of densely packed palisade cells, but the spongy parenchyma is absent [54]. The leaves of guayule are amphistomatous, and both surfaces are covered with dense trichomes, which are epidermal appendages forming a thick protective indumentum. The restriction of mesophyll tissue to palisade cells considerably increases the photosynthetically active surface of the leaf, which has less resistance to the diffusion of CO_2_ and can transmit a greater amount of PAR thanks to the optimised parallel arrangement of the palisade cells with the aligned chloroplasts [32]. As a result, even the abaxial palisade cells can receive sufficient light and contribute to the total leaf CO_2_ assimilation rate. In addition, the dense covering with trichomes protects the leaf against desiccation by reducing the cuticular water loss [42]. The physiological implications of these morphoanatomical characteristics in guayule can be improved and/or enhanced with the increased leaf thickness under drought or saline conditions [32]. Even in the presence of higher NaCl concentrations, the morphophysiological modifications observed here in the guayule leaves could result from adaptive mechanisms to salt stress aimed at maintaining a reasonable level of carbon fixation and organ development, despite the general loss in the surface and biomass. This is consistent with the smaller and thicker leaves formed in guayule under high-saline conditions, as well as with the stability of the LMR, NAR and most biometric parameters in the stem, roots and whole plant, including the stem/leaf and shoot/root ratios (Table 1). In this regard, particularly noteworthy is the maintenance of the stem length and its relative biomass allocation concerning the leaves and whole plant (unchanged SLMR and SMR), since it is in the stem that the rubber and resin will be produced [22,29].

The photosynthetic pigment composition of guayule leaves revealed that the changes in the total chlorophyll observed during the NaCl tolerance test corresponded to an effective reduction both in Chl *a* and Chl *b* at the end of the screening (40 g L^−1^ NaCl), associated with a significant decrease in carotenoids (Figure 6). One of the most common effects in plants under salt stress is chlorosis and the premature senescence of the leaves due to the reduced content of chlorophylls, and their variations are considered rapid biochemical indicators of salt tolerance [44,55]. The decreased content of leaf photosynthetic pigments can be due to the degradation of their biochemical structure and function, or to the inhibition of their synthesis. High salt concentrations in the medium can directly and indirectly provoke such events. Excess NaCl makes it harder for roots to uptake nutrients as, for example, Mg, essential in chlorophyll biosynthesis; the entry of Na^+^ in plants and its build-up to toxic levels within the cells can damage biomolecules and the structure by the disruption or substitution with other cations, such as Mg^2+^ in the chlorophylls, Ca^2+^ in the membranes and K^+^ in the catalytic activities of many enzymes [44,49,56]. Finally, the accumulation of disruptive biomolecules, nutrient imbalances and the consequent reduced rate of photosynthesis provoke oxidative stress, with further damage to photosynthetic pigments, including carotenoids and other compounds [51]. Carotenoids, in particular, play an important role in photosynthesis and photoprotection mechanisms, being accessory pigments and nonenzymatic scavengers of oxidative species [57,58]. Thus, the total amount of photosynthetic pigments is not only fundamental for the efficient functionality of the plant photosynthetic apparatus, but also the relative content of Chl *a*, Chl *b* and carotenoids and their combination in the two photosystems and related antenna complexes [47,57]. In the leaves of guayule, the highest NaCl addition of 40 g L^−1^ decreased the total chlorophyll and Chl *a*/*b* (as Chl *a* was more impaired than Chl *b*), whereas it did not affect the (Chl *a* + *b*)/carotenoids ratio. This could mean that the salt stress damaged both chlorophylls and carotenoids to the same extent, but not their relationships, compromising only partially the functionality of the photosystems. Another explanation might simply be that the number of efficient photosystems (and chloroplasts) was reduced.

A high NaCl concentration interferes with plant nutrition through the reduction in the root capacity in uptaking nutrients from the medium (osmotic effect) and through the competition and/or substitution of Na with other elements, such as K and Ca, inside the plant (ionic effect) [37,59]. In most cases, NaCl increases the Na concentration in plants while altering that of P, K, Mg, Ca and some micronutrients, e.g., Refs. [59,60,61]. In our experiment, the high NaCl levels applied did not change the P concentration in the leaves, stem and roots of guayule, revealing the ability of the plant to maintain the uptake of this macronutrient involved in the energy storage process, photosynthesis, root formation and growth [62]. In the same conditions, the K concentration was reduced in the roots and Ca in the stem of guayule. Both K^+^ and Ca^2+^ cations are competitors of Na^+^ under saline conditions in several plant processes, including osmotic adjustment, cellular signalling and the coordination of enzymatic activities as cofactors [63,64]. On the other hand, Mg increased in the stem, showing a higher uptake by the roots and/or remobilisation in the shoot of this element, which will probably be used in the synthesis of new chlorophyll molecules after their degradation caused by the excess Na [49,65]. These modifications of K, Ca and Mg observed in guayule under salt stress are consistent with a plant response mechanism mainly based on the tolerance of tissues in absorbing and accumulating Na instead of excluding it [37]. The Na partitioning with the other main cations showed its high influx and accumulation into the guayule tissues. The highest values of K/Na, Ca/Na and Mg/Na were found in the leaves, indicating a more elevated ability than in the stem and roots to maintain the balance of these nutrients despite the external Na concentrations [64]. Indeed, the concentrations of Na reached in the leaves, stems and roots of guayule exposed to 40 g L^−1^ NaCl were really elevated (about 40,000 mg kg^−1^ DW in all three organs) and comparable to the levels reached by many halophytes [9,66]. In guayule irrigated with saline waters (a maximum EC of 10 dS m^−1^), Placido et al. [27] found similar Na concentrations in the leaves (40,000–50,000 mg kg^−1^ DW), while Bãnuelos et al. [33] found lower but still high Na concentrations in the leaves (<21,000 mg kg^−1^ DW), stem (<10,000 kg^−1^ DW) and roots (<20,000 mg kg^−1^ DW). Such results supported the ability of guayule to tolerate high Na tissue concentrations. Moreover, in our experiment, the total amount of Na accumulated in different organs of guayule and removed by the whole plant (17.5 ± 2.0 mg Na plant^−1^) was relatively low if compared to the potential of halophytes, but very high for most glycophytic species [9,49]. However, the cell compartmentalisation of Na and its concentration capacity in different tissues of guayule was high, as shown by the BCF, which was one or almost one in leaves, stem and roots [67,68]. In addition, the ability of guayule to translocate the Na uptaken by the roots to the shoot was high (~8) and comparable to that of some halophytic plants [5,9,16]. The morphophysiological and biochemical parameters analysed in guayule showed that, in the vegetative phase of its first year of growth, this plant species is tolerant to high salt concentrations (15–20 g L^−1^ NaCl), levels at which only halophytic species can grow and complete their biological cycle [16,66]. In this phenological phase, guayule can survive in higher saline conditions close to hypersaline concentrations (35–40 g L^−1^), adopting strategies of tissue tolerance to the accumulated Na. In this context, guayule can be considered a halotolerant plant species. Moreover, an elevated amount of Na was found in the leaves compared to the stem and roots, and the leaves of guayule, containing negligible quantities of rubber, represent a waste byproduct in the industrial processes of rubber extraction [25].

The continuous rise in salinity inhibits plant growth and limits crop yield, with serious threats to agricultural production worldwide. Most of halophytic species, naturally tolerant to high salt levels (~200 mM or 12 g L^−1^ NaCl), are not cultivated plants. However, glycophytes are differently sensitive to saline conditions, and major cereal crops (barley, sorghum, wheat, triticale and oats) exhibit high tolerance to soil salinity [37,69]. The nonhalophytic plants, such as guayule originating from extreme or marginal environments, including arid or semiarid regions, are adapted to several environmental stressors (e.g., drought, intense sunlight, strong winds), and thus are promising resources for the selection of crops resistant to high-saline conditions. However, the previous works on guayule under saline conditions considered lower salt levels (EC of 10 dS m^−1^), and actually, there are no reports on this species subjected to higher NaCl treatments or comparable to those tested here. Still, a few other non-halophytic crop species have endured high levels of NaCl. Recently, in a field trial, Shao et al. [34] studied the effects of high salinity (7.23–8.15 g kg^−1^) on the tuber biomass and sugar metabolism of Jerusalem artichoke (*Helianthus tuberosus* L.), a perennial plant native to North America and belonging to the Asteraceae family, as is guayule. This salt level promoted the conversion of reducing sugars to nonreducing ones, but significantly decreased the tuber biomass of Jerusalem artichoke. Moreover, Ren et al. [35] tested the irrigation with brackish and saline waters on cotton (*Gossipium hirsutum*, L.), a moderately salt-tolerant crop (with a threshold level of 7.7 dS m^−1^) originating from Mexico, using increasing salinity levels from 1 to 12 g L^−1^. In this case, the irrigation with 3 g L^−1^ saline waters was beneficial for cotton growth, but a salinity level of more than 6 g L^−1^ inhibited the leaf expansion and dry mass. In guayule, although at a different growth stage, similar effects were found in the presence of 15 g L^−1^ NaCl. All these non-halophytic crops adapted to arid and semiarid areas are characterised by a high genetic variation within and among species, and genomic studies to investigate such variability and utilise tolerant genotypes are a prerequisite to the improvement of plant/crop selection for the phytomanagement and phytodesalinisation of salt-affected areas [69].

For the above reasons, nonfood plants or industrial crops such as guayule could be cultivated in marginal lands, such as saline environments, with the dual scope of reclaiming soil and water from excess salts and obtaining commercial products. In this framework, the cultivation and uses of guayule fall well within the initiatives of the 17 UN Sustainable Development Goals (SDGs) adopted by all the United Nations Member States in 2015 to design new generations of bio-based products obtained with sustainable processes [22].

## 4. Materials and Methods

### 4.1. Plant Material, Preculture, Growth Conditions and Treatments

Guayule (*Parthenium argentatum* A. Gray) was obtained from a commercial nursery in southern Italy (37°05′41.5″ N, 14°14′30.0″ E), where the plants were generated from seeds and grown inside phytocells filled with organic soil. Seedlings, about two months old, were acclimated to greenhouse conditions: a 16–30 °C daytime temperature range, a 50–60% relative humidity, 180 ± 40 µmol m^−2^ s^−1^ of photosynthetic active radiation (PAR, 10.00 a.m.–5.00 p.m.) and a 430 ± 60 µmol mol^−1^ CO_2_ concentration. The greenhouse was located in the 123,000 m^2^ campus of the National Research Centre (CNR), Pisa (43°43′10.0″ N, 10° 25′11.5″ E). Guayule was gradually adapted to hydroponic conditions: the soil was carefully removed from the roots, which were thoroughly washed with running tap water, and then each plant was placed in backers containing a liquid commercial fertiliser diluted in tap water for about two weeks. Then, well-grown and homogeneous plants were selected and transferred to a hydroponic floating root system consisting of separate rectangular containers (19 cm height × 15 cm width × 26 cm length), each filled with 3 L of hydroponic solution and containing three plants. The hydroponic nutrient solution was prepared with a commercial complete liquid fertiliser system (FLORA Series, General Hydroponics Europe—GHE, France) diluted in tap water. This nutrient solution was the combination of three fertiliser components: FloraGro 2-1-6, FloraMicro 5-0-1 and FloraBloom 0-5-4. FloraGro contains: 2% total nitrogen (N), 1% phosphoric anhydride (P_2_O_5_), 6% potassium oxide (K_2_O) and 0.8% magnesium oxide (MgO). FloraMicro: 5% N, 1% K_2_O, 6% calcium oxide (CaO), 0.01% boron (B), 0.01% copper (Cu)-chelated EDTA, 6% iron (Fe)-chelated (EDDHA-11% and DPTA-0.12%), 0.07% manganese (Mn)-chelated EDTA, 0.002% molybdenum (Mo) and 0.02% zinc (Zn)-chelated EDTA. FloraBloom: 5% P_2_O_5_, 4% K_2_O, 3% MgO and 5% sulphuric anhydride (SO_3_). A polystyrene plate was placed on the edge of each container to provide physical support for the plants and reduce the solution’s evaporation. At the root collar point, each plant was inserted into a special hole in the polystyrene plate, where a plastic support kept the plant erect; the solution was kept constantly aerated by aquarium pumps (250 L h^−1^). After acclimation to the hydroponics, the plants (about 3.5 months old) were randomly distributed to the two different experimental growth conditions: control plants supplied with the basal nutrient solution and treated plants supplied with the basal nutrient solution added with amounts of sodium chloride (NaCl) to reach the concentration at each time tested (Figure 1). The tested NaCl concentrations of the treatments were set at 2.5, 5.0, 10.0, 15.0, 20.0, 25.0, 30.0 and 40.0 g L^−1^, corresponding to 42.8, 85.6, 171.1, 256.7, 342.2, 427.8, 513.3 and 684.4 mM NaCl, respectively. These NaCl concentrations were selected based on a previous salt tolerance screening on guayule. The NaCl addition was carried out gradually to avoid osmotic shock and increased every 3–5 days. The nutrient solution was renewed twice a week, and the pH was adjusted to 5.5–6.5 (Appendix A) to maintain the optimal nutrient supply for plant growth and, in the treatments, to maintain the stable ratio between sodium (Na) and the other cations, especially calcium (Ca, Na/Ca). The electrical conductivity (EC) and pH of the control and NaCl-treated solutions were measured at least twice a week with a portable instrument (2301T conductivity meter and digital pH electrode, XS Instruments, Carpi, MO, Italy). The EC of the solutions added with NaCl varied from 6.6 to 64.5 dS m^−1^ (Appendix A). The experimental test duration and growth time of the plants was 30 days. Two containers with three plants each were used for the controls and two for the NaCl treatments, so there were six biological replicates or plants (*n* = 6) per treatment (Figure 1).

### 4.2. Monitoring through Nondestructive Measurements

#### 4.2.1. Biometric Parameters and Total Chlorophyll

From the setup of the experimental trial (t_0_) and the beginning of the NaCl tolerance test (t_1_), the maximum length of the root apparatus and the number of leaves of each guayule plant (control and treated) were registered every 2–3 days until the end of experiments (t_30_). Simultaneously, the maximum length (L) and width (W) of the leaves (main orthogonal leaf diameters) were measured, and the eventual visual symptoms of stress (deformation, yellowing or browning of leaves, epinasty or falling) were observed. Moreover, the same leaf area (about 1.0 cm diameter; that is, 0.7854 cm^2^ in area) from the same guayule plant used for the chlorophyll fluorescence and chemical determinations (Section 4.2.2 and Section 4.3.2) was analysed for the estimation of total chlorophyll with a portable chlorophyll meter (SPAD-502Plus meter, Konica-Minolta, Inch., Osaka, Japan). Completely expanded leaves were chosen for such measurements, covering a wide range of vital and differently aged leaves and avoiding the mid-rib. As light intensity can affect the light transmittance of the leaves [70], the SPAD readings were always taken at the same time of day.

#### 4.2.2. Chlorophyll Fluorescence

Chlorophyll fluorescence measurements were conducted on guayule plants to evaluate the NaCl effects on the functionality of the photosystem II (PSII). Measurements were conducted in correspondence with the salt solution renewal (twice a week) on intact, fully expanded and exposed leaves using a pulse-amplitude-modulated fluorometer (Mini-PAM; Heinz Walz GmbH, Effeltrich, Germany) on each experimental plant.

The actual photon yield of the PSII in the light (Φ_PSII_) was measured on long-term light-adapted leaves (180 ± 40 µmol m^−2^ s^−1^ of PAR, growing light conditions) and determined as Φ_PSII_ = (Fm′ − Fs)/Fm′ at steady state, where Fm′ represents the maximum fluorescence yield with all the PSII reaction centres in the reduced state obtained by superimposing a saturating light flash during the exposure to actinic light, and Fs is the fluorescence at the actual state of the PSII reaction centres during actinic illumination [71].

The potential efficiency of the PSII photochemistry was evaluated on plants adapted to dark for at least 30 min as Fv/Fm = (Fm − Fo)/Fm, where Fv is the variable fluorescence in the dark, Fo represents the minimum fluorescence yield in the dark and Fm is the maximum fluorescence yield in the dark after the application of the saturation flash of light, which completely closes all the PSII reaction centres [71,72].

#### 4.2.3. Gas-Exchange Parameters

Based on the results of the chlorophyll fluorescence measurements, gas-exchange parameters were determined to evaluate the specific plant response at 15 g L^−1^ NaCl. Measurements were performed on each plant at 10.00–12.00 a.m. on the same leaves used for the chlorophyll fluorescence with a photosynthetic portable system Li6400 (LiCor, Lincoln, NE, USA). Chamber conditions were set at 20 °C and 400 µmol mol^−1^ of the CO_2_ concentration. The PPFD (photosynthetic photon flux density) inside the chamber was set at growing light conditions (180 µmol mol^−2^ s^−1^) with a red/blue LED light source. The evaluated parameters were the steady-state photosynthetic CO_2_ assimilation rate (A), stomatal conductance (gs), intercellular CO_2_ concentration (Ci) and transpiration rate (E). The intrinsic and instantaneous water-use efficiencies (WUEs) were calculated as A/gs and A/E, respectively [73].

### 4.3. Analysis through Destructive Measurements

#### 4.3.1. Growth, Hydration Status, Biomass Partitioning and Analysis

Destructive measurements were performed before the beginning (t_0_) and at the end (t_30_) of the NaCl tolerance test (Figure 1). Guayule plants were harvested and separated into the leaves, stems and roots; the fresh weight (FW) of this plant material was immediately recorded. The length of the stem (SL) and roots (RL) of each plant was measured, and single leaves were scanned for the determination of the leaf area (LA).

Leaf discs (Ø, 7 or 10 mm) from at least six completely expanded leaves of three different plants for each treatment were carefully sampled (avoiding the mid-rib) with a cork borer and used for the determination of the leaf relative water content (RWC). The leaf discs were immediately weighed to obtain the sample FW; then, they were hydrated to full turgidity for one night by floating on deionised water in a closed Petri dish at 4 °C. After hydration, the leaf discs were removed from the water, any surface moisture was quickly dried off using tissue paper and weighed to obtain the fully turgid weight (TW). Finally, the leaf disks were oven-dried at 50 °C until a constant weight and weighed to determine the dried weight (DW). The leaf RWC was calculated from the following Equation (1):RWC (%) = [(FW − DW)/(TW − DW)] × 100(1)

After the length measurements and scans, the recording of the FW and the collection of the leaf discs, all plant material was oven-dried at 50 °C to a constant weight for the determination of the total biomass DW and partitioning among the organs.

The primary data (weights, areas, lengths) obtained from guayule plants throughout the experiment were used to investigate the plant growth analysis [52,74] in response to the NaCl stress. The relative growth rate (RGR) was estimated from the slope of the linear regression according to Equation (2):ln Mt_30_ = ln Mt_0_ + RGR,(2)
where Mt_30_ is the total plant DW measured at the end of the experimental tests (i.e., 30 days of treatments at increasing NaCl concentrations) and Mt_0_ is the total plant DW immediately before the first salt treatment.

The leaf area ratio (LAR) was calculated using Equation (3):LAR = 1/LMA × LMR,(3)
where LMA is the leaf mass per area (leaf DW (g)/leaf area (m^2^)) and LMR, or the leaf mass ratio, is the fraction of the total biomass allocated to the leaves (leaf DW (kg)/plant DW (kg)).

The net assimilation rate (NAR), or the DW increment per leaf area unit, was calculated as in Equation (4):NAR = RGR/LAR,(4)

The length of the stem (corresponding to the plant height) and the major length of the root apparatus were used to calculate a series of parameters related to the plant growth analysis: SLMR, stem length mass ratio (SL (cm)/stem DW (g)); SMR, stem mass ratio (stem DW (g)/plant DW (g)) [74]; RLMR, root length mass ratio (RL (cm)/root DW (g)); RMR, root mass ratio (root DW (g)/plant DW (g)) [67,75].

#### 4.3.2. Extraction of Photosynthetic Pigments

Before the beginning (t_0_) and at the end (t_30_) of the NaCl tolerance test, in correspondence with the same leaf area covered by the emitting window of the SPAD device (Section 4.2.1), leaf discs (10 mm or 7 mm diameter: 0.785 cm^2^ or 0.385 cm^2^ area, respectively) were collected from each plant with a cork borer. These discs were immediately weighed and homogenised in 80% (*v*/*v*) cold acetone with the help of mortar and pestle. The extracts were centrifuged at 12,000 rpm for 10 min at 4 °C, and the absorbance of supernatants that was obtained, and eventually filtered (0.2 µm, Sartorius Stedim Biotech, Göttingen, Germany), was measured at 663.2, 646.8 and 470.0 nm using a UV–vis spectrophotometer (UV-1800 Spectrophotometer, Shimadzu, Kyoto, Japan). The concentration of photosynthetic pigments (chlorophyll *a*, Chl *a*; chlorophyll *b*, Chl *b*; total carotenoids) was performed using the equations indicated by [76] and expressed as µg on a leaf area basis (cm^2^).

#### 4.3.3. Chemical Organ Analysis: Distribution of Sodium and Nutrients

In addition to sodium (Na), the nutrients phosphorous (P), potassium (K), calcium (Ca) and magnesium (Mg) were determined in the leaves, stems and roots of guayule plants for each experimental condition according to Barbafieri et al. [9]. The oven-dried plant material was ground to a fine powder. Well-homogeneous samples (0.5 g) were digested in a solution of nitric acid (HNO_3_ 65%) and hydrogen peroxide (H_2_O_2_ 30%) (2:1, *v*/*v*) in a closed-vessel microwave-assisted digestion system (Milestone Ethos 900, Bergamo, Italy) using US-EPA Method 3052. The content of K, Mg and Ca was analysed by inductively coupled plasma–optical emission spectroscopy (ICP-OES 5900, Agilent, Santa Clara, CA, USA), and the Na concentration was determined using atomic absorption spectrometry (AA240 FS, Agilent Technologies, Santa Clara, CA, USA). The content of elemental P was performed following the molybdate blue ascorbic acid method [77].

#### 4.3.4. Sodium Translocation, Accumulation and Concentration Factors

The capacity of guayule to uptake Na, transfer it to the shoot and, eventually, accumulate in some organs was evaluated with the investigation of Na contents in the leaves, stems and roots, as well as with the calculation of the bioconcentration (BCF) and translocation (TF) factors [67,68] at the end of the experiment (t_30_).

The sodium content or uptake in different plant organs was determined by multiplying the Na concentration (mg g^−1^ DW) per DW (g) of the corresponding biomass. The BCF and TF were calculated according to the following Equations (5) and (6):BCF = [Na]_plant organ_/[Na]_medium_,(5)
TF = Na_shoot_/Na_roots_,(6)
where [Na]_plant organ_ and [Na]_medium_ represent the concentration (g kg^−1^) of Na in the plant organ considered (leaves, stems or roots) and in the nutrient solution (g L^−1^); Na_shoot_ and Na_roots_ are the content (mg) of sodium in the shoots and roots. The BCF and TF higher than 1 indicate a high plant capability for the Na tissue concentration and Na translocation from the roots to shoots, respectively.

### 4.4. Regression Models for the Leaf Area and Total Chlorophyll Determinations through Nondestructive Measurements

The leaf nondestructive morphophysiological measurements (maximum length–width and SPAD values) recorded throughout the NaCl tolerance test (Section 4.2.1) were related to the destructive determinations of the leaf area (LA) and total chlorophylls (Chl *a* + Chl *b*) performed before (t_0_) and at the end (t_30_) of the NaCl treatments. The derived relation models aim to monitor the plant behaviour during the salt treatments without defoliation and the consequent loss of biomass with the samplings. The LA was measured using a scanner acquisition system and an imaging analysis software (ImageJ, IJ 1.46r, http://imagej.nih.gov/ij/, accessed on 7 November 2020, Section 4.3.1). Total chlorophylls were obtained by the chemical extraction of Chl *a* and Chl *b* from the leaf discs (Section 4.3.2) collected in correspondence with the same leaf area used for the SPAD measurements (Section 4.2.1).

The LA measurements obtained by the scans and image analysis were used to create correlations with the leaf morphological nondestructive determinations, such as the length and width. After some attempts, we found the best relationship between the LA and the product of the leaf length (L) and width (W); that is, the two main orthogonal leaf diameters (L × W). The relationships between the two datasets (LA and (L × W)) were analysed using both R Studio (2020) and Excel (Microsoft, 2019) software [74]. Several models were tested, and the two packages provided identical results, indicating that linear functions provide the best fit (*p* < 0.0001): LA = b + a (L × W), where “a” is the slope and “b” is the intercept of the line. As salt stress may cause leaf morphological changes, the analysis of the covariance (ANCOVA, *p* ≤ 0.05) was applied to verify if the linear regression between the LA and (L × W) of the controls was significantly different from that of the NaCl-treated plants. Results showed that neither the slope nor the intercept of the two linear regressions for the control and treated plants were significantly different, with *p* = 0.646 and *p* = 0.240, respectively (Appendix A).

The leaf Chl *a* and Chl *b* extractions from the controls and treated plants at t_0_ and t_30_ were used to create correlations with the leaf SPAD total chlorophyll estimation during the NaCl tolerance test. Also, in this case, the linear regression provided the best fit (*p* < 0.0001) between the two datasets: (Chl *a* + *b*) = b1 + a1 (SPAD). Results of the ANCOVA (*p* ≤ 0.05) to compare the regression linear curves of the guayule controls and the NaCl-treated plants showed that the slope of the regression line was different between the controls and the treated plants (*p* = 0.012), indicating a significant interaction between the two experimental conditions (Appendix A).

### 4.5. Statistical Analysis

The experiments were set up in a completely randomised design with six replicate plants (*n* = 6) for each treatment. One-way analysis of variance (ANOVA, *p* ≤ 0.05) was applied to evaluate the effect of the increasing sodium (Na) concentrations on guayule at each NaCl addition time. Data shown in graphs and tables represent the mean values ± standard error (SE). Separation of means was performed using Fisher’s least significance difference (LSD) test at a significance level of *p* ≤ 0.05. To evaluate the significance of the regression model fitting data and to compare the eventual differences between the linear regressions, the analysis of covariance (ANCOVA) was used. For the percentage results, before the statistical analysis, an arcsine transformation was applied. Statistical analyses were conducted using STATISTICA software (version 7.0, Stat-Soft, Inc., Tulsa, OK, USA), R Studio (2020) and Microsoft Excel (2019).

## 5. Conclusions

Guayule is a perennial shrub naturally adapted to arid climates and recently domesticated to produce rubber and resins. Although resistant to drought, this industrial crop must be irrigated for maximum sustained production, and this practice, in arid and semiarid regions, can create serious concerns for salinity, aggravated by climate change. We investigated the response of guayule to high and increasing saline conditions. Results showed that guayule can adapt to high salt concentrations (15–20 g L^−1^ or 257–342 mM NaCl), showing mechanisms of osmotic and tissue tolerance to Na stress. This species can survive in the presence of NaCl levels close to hypersaline conditions (35–40 g L^−1^ NaCl), but its growth and morphophysiological traits were strongly affected. The capacity of guayule to absorb and accumulate high Na levels in its tissues in the presence of extreme saline conditions was shown by the elevated translocation and bioconcentraction factors.

To the best of our knowledge, this is the first study on the response of guayule to high and increasing concentrations of NaCl with the aim of evaluating the plant morphophysiological response, its tolerance threshold to salt and the capacity of accumulating Na in different organs.

This study contributes to identifying or selecting glycophytes/halotolerant plants able to survive, thrive and produce in high-saline environments; guayule could be used in the phytomanagement and/or phytodesalinisation of saline waters in constructed wetlands and the remediation of salt-rich or contaminated soils. To better understand the mechanisms underlying the tolerance capacity to the high salt stress of the drought-adapted shrub guayule, comprehensive genomic experiments are needed, and they will be one of next steps of our research.

## Figures and Tables

**Figure 1 plants-13-00378-f001:**
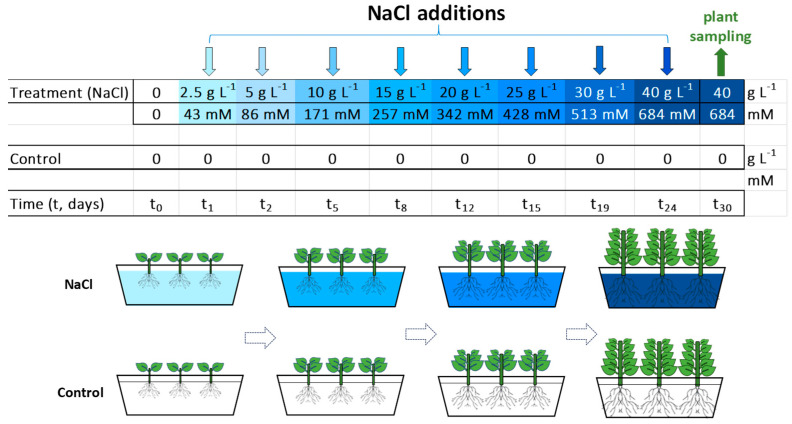
Scheme of the experimental setup and time-course of the NaCl screening test in the hydroponic floating root system. Guayule plants (*Parthenium argentatum* A. Gray), about 3.5 months old, were exposed to the increasing NaCl concentrations shown (from 2.5 to 40 g L^−1^; that is, from 42.8 to 684.4 mM). Each hydroponic unit (separate rectangular container) included three plants and two units for every treatment (control and NaCl) were used. The single plant was the biological replicate; then, there were six replicates per treatment (*n* = 6), and they were grown in these conditions for 30 days.

**Figure 2 plants-13-00378-f002:**
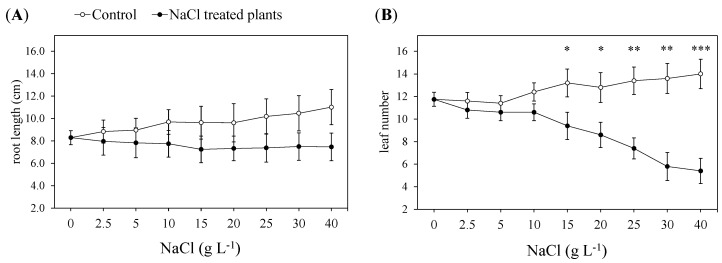
Maximum root length (**A**) and leaf number (**B**) of *P*. *argentatum* exposed to increasing NaCl concentrations (from 2.5 to 40 g L^−1^; that is, from 42.8 to 684.4 mM). Values are means ± standard error (SE) of six biological replicates or plants (*n* = 6). The white circles represent the control plants (0 NaCl) and the black circles represent the NaCl-treated plants. Values marked with an asterisk or asterisks corresponding to each NaCl treatment are significantly different compared to the control at *p* ≤ 0.05 (*), *p* ≤ 0.01 (**) and *p* ≤ 0.001 (***) according to a one-way ANOVA. Values not marked with asterisks are not significantly different compared to the control (*p* > 0.05).

**Figure 3 plants-13-00378-f003:**
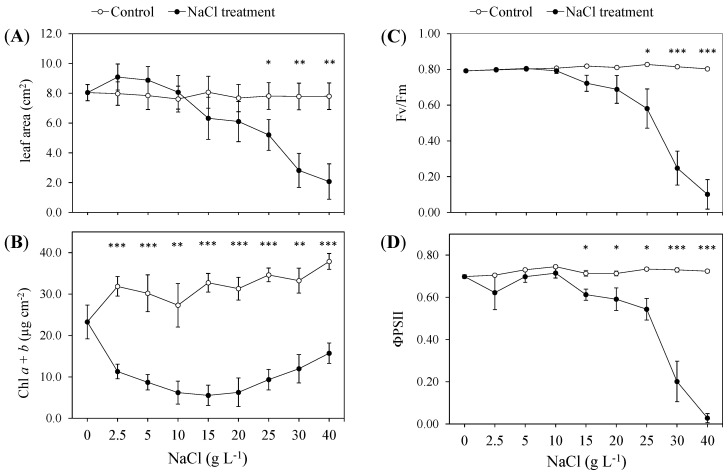
Mean area (**A**), total chlorophyll (Chl *a* + *b*) concentration (**B**), maximum (Fv/Fm, (**C**)) and actual (ΦPSII, (**D**)) quantum yield of the photosystem II (PSII) in the leaves of *P*. *argentatum* exposed to increasing NaCl concentrations (from 2.5 to 40 g L^−1^; that is, from 42.8 to 684.4 mM). Values are means ± standard error (SE) of six biological replicates or plants (*n* = 6). The leaf Chl *a* + *b* during the test was evaluated thanks to a regression model between the measurements with a portable Chl meter (SPAD-502Plus meter, Konica-Minolta, Osaka, Japan) and the chemical extraction of photosynthetic pigments at the beginning and end of the NaCl treatments. The white circles represent the control plants (0 NaCl) and the black circles represent the NaCl-treated plants. Statistics and symbols as in Figure 2.

**Figure 4 plants-13-00378-f004:**
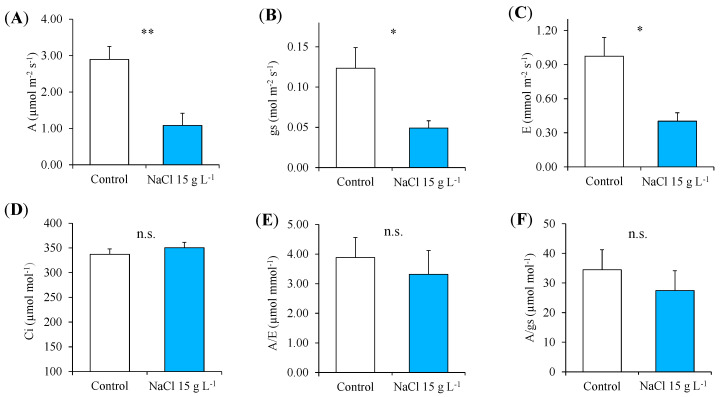
Leaf–gas exchanges in the control (0 NaCl) and the treated (15 g L^−1^ NaCl) plants of guayule (*P*. *argentatum*). (**A**) CO_2_ assimilation rate, A; (**B**) stomatal conductance, gs; (**C**) transpiration rate, E; (**D**) intercellular CO_2_ concentration, Ci; (**E**) instantaneous water-use efficiency, A/E; (**F**) intrinsic water use efficiency, A/gs. Values are means ± standard error (SE) of six biological replicates or plants (*n* = 6). One-way ANOVA (*p* ≤ 0.05) was applied to evaluate the effect of the NaCl treatment. n.s.: not significant; * and **: significantly different at *p* ≤ 0.05 and *p* ≤ 0.01, respectively.

**Figure 5 plants-13-00378-f005:**
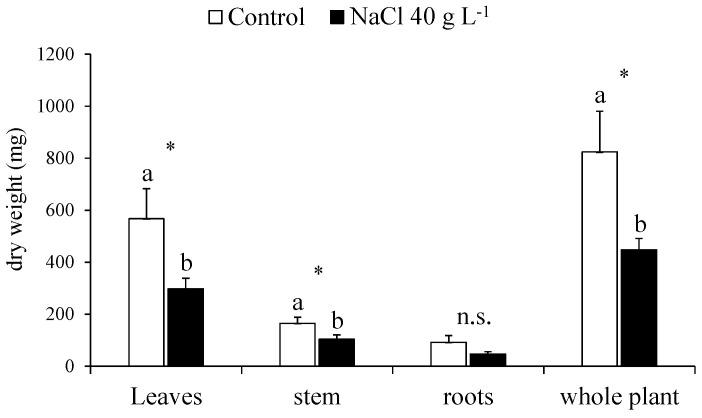
Whole-plant biomass production and partitioning in the leaves, stem and roots in the control (0 NaCl) and treated (40 g L^−1^ NaCl) plants of guayule after 30 days of growth under increasing NaCl concentrations. Values are means ± standard error (SE) of six biological replicates or plants (*n* = 6). Data were analysed independently by a one-way ANOVA (*p* ≤ 0.05) to evaluate the effect of the NaCl treatment. Mean values indicated with different letters are significantly different, according to Fisher’s LSD. n.s., not significant; *, significantly different at *p* ≤ 0.05.

**Figure 6 plants-13-00378-f006:**
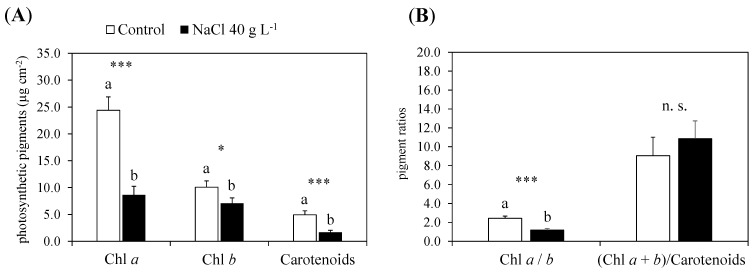
Leaf concentration of chlorophylls and carotenoids (**A**) and pigment ratios (**B**) in the control (0 g L^−1^ NaCl) and treated (40 g L^−1^ NaCl) plants of guayule after 30 days of growth under increasing NaCl concentrations. Values are means ± standard error (SE) of six biological replicates or plants (*n* = 6). Statistics as in Figure 5.; n.s., not significant; * and ***, significantly different at *p* ≤ 0.05 and *p* ≤ 0.001, respectively. Abbreviations: Chl *a*, chlorophyll *a*; Chl *b*, chlorophyll *b*; Chl *a* + *b*, total chlorophyll.

**Figure 7 plants-13-00378-f007:**
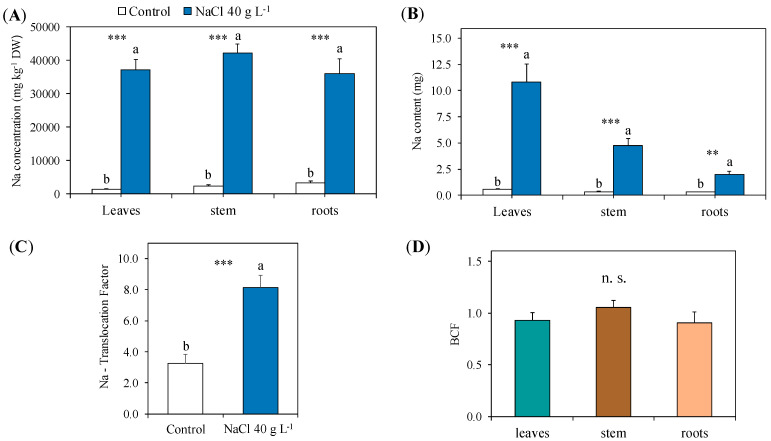
Sodium (Na) concentration ((**A**), mg g^−1^), content ((**B**), mg), translocation factor ((**C**), TF) and bioconcentration factor ((**D**), BCF) in different the organs of the control (0 NaCl g L^−1^) and treated plants of guayule exposed for 30 days to increasing concentrations of NaCl up to 40 g L^−1^. Values are means ± SE (*n* = 4). Statistics as in Figure 5; n.s., not significant; ** and ***, significantly different at *p* ≤ 0.01 and *p* ≤ 0.001, respectively.

**Table 1 plants-13-00378-t001:** Growth parameters and analysis of guayule (*P*. *argentatum* A. Gray) exposed for 30 days to increasing concentrations of NaCl up to 40 g L^−1^ (Tr., treated). Control plants (Ctr.) were grown without (0 g L^−1^) NaCl. Values are means ± standard error (SE) of six biological replicates or plants (*n* = 6). Data were analysed independently by a one-way ANOVA (*p* ≤ 0.05). Mean values followed by different letters in the same row are significantly different according to Fisher’s LSD test. The Fisher value (F) and significance level (*p*, in italics, when ≤ 0.050) are shown.

Growth Parameter	NaCl Treatment		Statistics	
leaf	Ctr.	Tr.	F	*p*
FW/DW	4.28 ± 0.43 a	1.28 ± 0.11 b	45.07	*<0.001*
RWC (%)	78.75 ± 4.96 a	28.94 ± 11.03 b	20.89	*0.001*
LMA (g m^−2^)	56.43 ± 1.29 b	145.23 ± 12.95 a	46.57	*0.002*
LAR (m^2^ kg^−1^)	12.03 ± 0.37 a	4.13 ± 0.50 b	164.28	*<0.001*
LMR (g g^−1^)	0.67 ± 0.02	0.66 ± 0.05	0.01	0.922
NAR (g m^−2^ d^−1^)	3.39 ± 0.58	3.32 ± 1.64	0.002	0.969
stem	Ctr.	Tr.		
FW/DW	3.07 ± 0.14 a	2.46 ± 0.18 b	7.13	*0.028*
SL (cm)	4.88 ± 0.23	4.66 ± 0.45	0.19	0.672
SLMR (cm g^−1^)	31.59 ± 2.21	40.23 ± 6.23	2.06	0.194
SMR (g g^−1^)	0.23 ± 0.03	0.24 ± 0.04	0.046	0.835
stem/leaf	0.35 ± 0.05	0.40 ± 0.12	0.16	0.697
root	Control	Tr.		
FW/DW	9.35 ± 0.77	7.54 ± 0.78	0.88	0.377
RL (cm)	9.49 ± 1.32	9.00 ± 1.33	0.068	0.801
RLMR (cm g^−1^)	132.94 ± 16.92	157.68 ± 23.11	0.69	0.433
RMR (g g^−1^)	0.11 ± 0.02	0.10 ± 0.02	0.049	0.830
Whole plant	Ctr.	Tr.		
LA (cm^2^)	87.65 ± 25.40 a	16.02 ± 2.61 b	7.87	*0.049*
FW/DW	4.49 ± 0.46 a	2.14 ± 0.11 b	24.28	*0.001*
RGR (d^−1^)	4.00 × 10^−2^ ± 5.75 × 10^−3^ a	2.44 × 10^−2^ ± 3.11 × 10^−3^ b	5.66	*0.044*
shoot/root	10.02 ± 2.94	10.65 ± 3.22	0.021	0.889

FW, fresh weight; DW, dry weight; RWC, relative water content; LMA, leaf mass per area; LAR, leaf area ratio; LMR, leaf mass ratio; NAR, net assimilation rate; SL, stem length; SLMR, stem length mass ratio; SMR, stem mass ratio; RL, root length; RLMR, root length mass ratio; RMR, root mass ratio; LA, leaf area; RGR, relative growth rate.

**Table 2 plants-13-00378-t002:** Concentration (mg g^−1^ DW) of potassium (K), phosphorous (P), calcium (Ca) and magnesium (Mg) in different organs (leaves, stem and roots) of guayule, control (Ctr., 0 NaCl g L^−1^) and treated (Tr.) plants exposed for 30 days to increasing concentrations of NaCl up to 40 g L^−1^. Values are means ± SE of four biological replicates or plants (*n* = 4). Mineral content was analysed by a one-way ANOVA (*p* ≤ 0.05). Statistics as in Table 1.

Element	Plant Organ	NaCl Treatment		Statistics	
		Ctr.	Tr.	F	*p*
K (mg g^−1^)	leaves	36.51 ± 4.69	30.22 ± 2.89	1.31	0.297
	stem	16.46 ± 2.24	12.06 ± 2.19	1.98	0.209
	roots	31.82 ± 5.03 a	10.21 ± 1.31 b	17.26	*0.006*
P (mg g^−1^)	leaves	1.84 ± 0.57	1.61 ± 0.06	0.16	0.705
	stem	4.10 ± 1.17	5.77 ± 0.61	1.61	0.251
	roots	7.47 ± 0.91	9.67 ± 2.71	0.59	0.471
Ca (mg g^−1^)	leaves	17.32 ± 0.73	18.20 ± 2.35	0.13	0.734
	stem	11.12 ± 1.30 a	5.39 ± 1.84 b	6.47	*0.044*
	roots	7.81 ± 2.39	9.13 ± 1.59	0.21	0.662
Mg (mg g^−1^)	leaves	4.05 ± 0.16	4.85 ± 0.68	1.30	0.297
	stem	0.49 ± 0.08 b	1.14 ± 0.04 a	49.05	*<0.001*
	roots	1.44 ± 0.23	1.30 ± 0.08	0.33	0.588

**Table 3 plants-13-00378-t003:** Ratio of the major cations (potassium, K; calcium, Ca; magnesium, Mg) with sodium (Na) in the different organs of the guayule, control (Ctr., 0 NaCl g L^−1^) and treated (Tr.) plants exposed for 30 days to increasing concentrations of NaCl up to 40 g L^−1^. Values are means ± SE of four biological replicates or plants (*n* = 4). Data were analysed by a one-way ANOVA (*p* ≤ 0.05). Statistics as in Table 1.

	NaCl Treatment		Statistics	
K/Na	Ctr.	Tr.	F	*p*
leaves	34.02 ± 11.01 a	0.84 ± 0.13 b	9.09	*0.024*
stem	8.31 ± 2.37 a	0.29 ± 0.05 b	11.40	*0.015*
roots	10.75 ± 3.72 a	0.29 ± 0.03 b	7.90	*0.031*
Ca/Na	Ctr.	Tr.		
leaves	15.09 ± 3.64 a	0.51 ± 0.10 b	16.01	*0.007*
stem	5.36 ± 1.21 a	0.13 ± 0.04 b	18.55	*0.005*
roots	2.09 ± 0.49 a	0.26 ± 0.05 b	13.71	*0.010*
Mg/Na	Ctr.	Tr.		
leaves	3.44 ± 0.69 a	0.14 ± 0.03 b	22.97	*0.003*
stem	0.26 ± 0.10 a	0.027 ± 0.002 b	5.42	*0.050*
roots	0.42 ± 0.04 a	0.04 ± 0.01 b	96.12	*<0.001*

## Data Availability

The data presented in this study are available on request from the corresponding authors. The data are not publicly available due to the restriction policy of the co-authors’ affiliations.

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
