# Peer review of "Morphophysiological Characterisation of Guayule (Parthenium argentatum A. Gray) in Response to Increasing NaCl Concentrations: Phytomanagement and Phytodesalinisation in Arid and Semiarid Areas"

_plants, 2024, doi:10.3390/plants13030378_

Round 1
Reviewer 1 Report
Comments and Suggestions for Authors
Some minor comments have been presented in the attached file.

Comments on the Quality of English LanguageExtensive editing of the English language required
Author Response
Dear reviewer 01,
Thank you for taking the time to review our manuscript.
The comments and suggestions permitted us to improve and better focalised our manuscript. We carefully considered and provided point-by-point responses to all your comments, corrections and suggestions. Please, see the attachment.
We also provided a new file of the manuscript with all the changes made marked up using the “Track Changes” function in MS Word, so that all modifications can be easily viewed by the reviewers and Editor.
We hope we have satisfied all your questions and suggestions and addressed all the issues raised.
Best regards,
Daniela Di Baccio.

Reviewer 2 Report
Comments and Suggestions for Authors
The author did a nice work to investigate for research on morpho-physiological and (bio)chemical response of guayule to different high and increasing levels of NaCl, and they showed parameters and indices useful for identifying its salt tolerance threshold, adaptative mechanisms and reclamation potential in high saline environments. This research is very important for agriculture. I'm wonder the author could do some compare on other reports for other species also could endure the high level of NaCl. Moreover, some genomic experiment could be added in this research for investigate the real mechanism.
Comments on the Quality of English LanguageThe author should need to revise the manuscript with the help of a Native English speaker, some mistakes in the gramma.
Author Response
Dear reviewer 02,
Thank you for taking the time to review our manuscript.
The comments and suggestions permitted us to improve and better focalised our manuscript. We carefully considered and provided point-by-point responses to all your comments, corrections and suggestions. Please, see the attachment.
We also provided a new file of the manuscript with all the changes made marked up using the “Track Changes” function in MS Word, so that all modifications can be easily viewed by the reviewers and Editor.
We hope we have satisfied all your questions and suggestions and addressed all the issues raised.
Best regards,
Daniela Di Baccio.

Reviewer 3 Report
Comments and Suggestions for Authors
The authors investigated the effects of increasing concentrations of NaCl on growth, morpho-physiological and biochemical characteristics of guayule, and aimed to evaluate its tolerance to salt stress and suitability in phyto-management and phyto-desalinization of salt-affected areas. The manuscript is well written, and the logic is fluent. Some minor revisions are needed below.
1) Notice the uppercase in the unit for [NaCl] in figures 4-7. g L-1 ->
2) Most of the semicolons can be replaced with periods. e.g., those at line 37, 149, 173, 219, 648…
3) Format should be taken care. E.g., 647 according to [9]-> according to Barbafieri et al. [9]
Author Response
Dear reviewer 03,
Thank you for taking the time to review our manuscript.
The comments and suggestions permitted us to improve and better focalised our manuscript. We carefully considered and provided point-by-point responses to all your comments, corrections and suggestions. Please, see the attachment.
We also provided a new file of the manuscript with all the changes made marked up using the “Track Changes” function in MS Word, so that all modifications can be easily viewed by the reviewers and Editor.
We hope we have satisfied all your questions and suggestions and addressed all the issues raised.
Best regards,
Daniela Di Baccio.

Reviewer 4 Report
Comments and Suggestions for Authors
Comments and Suggestions for Authors
The current study entitled “Morphophysiological characterization of guayule (Parthenium argentatum A. Gray) in response to increasing NaCl concentrations: phyto management and phytodesalinization of arid and semi-arid areas” investigated the effects of high and increasing concentrations of sodium chloride (NaCl) on growth, morphophysiological and biochemical characteristics of guayule, in order to evaluate its tolerance to salt stress and suitability in phyto management and, eventually, phytodesalinization of salt-affected areas.
Overall, the paper has been well written with adequate information in introduction, results, discussion, and well-described material and methods. There are only few comments:
Abstract
The abstract provides a clear objective of the study; however, it would be beneficial to add the hypotheses of the research in one or two sentences.
Keywords: The current keywords are phrases; I would suggest replacing them with words (three to ten keywords).
Introduction
The introduction is well written, but it is better to bring the result of other research on Guayule or similar species that are exposed to environmental stresses and compare the current research with previous ones, it could Improve this gap and also highlight the novelty of your study (in Paragraph 5).
Results and discussion and conclusion
These sections are well written.
Material and methods
Please add the latitude and longitude of the nursery (Lines 515-516) and also the greenhouse (line 518).
If the basic soil characteristics exist, I would suggest adding this information in section 4.1.
Author Response
Dear reviewer 04,
Thank you for taking the time to review our manuscript.
The comments and suggestions permitted us to improve and better focalised our manuscript. We carefully considered and provided point-by-point responses to all your comments, corrections and suggestions. Please, see the attachment.
We also provided a new file of the manuscript with all the changes made marked up using the “Track Changes” function in MS Word, so that all modifications can be easily viewed by the reviewers and Editor.
We hope we have satisfied all your questions and suggestions and addressed all the issues raised.
Best regards,
Daniela Di Baccio.

Reviewer 5 Report
Comments and Suggestions for Authors
Minor text issues:
line 72: i.e.
line 764: link of DOI underlined and in blue (probable text as link).
line 775: link of DOI underlined and in blue (probable text as link)
line 780: link of DOI underlined and in blue (probable text as link)
line 799: link of DOI underlined and in blue (probable text as link)
line 831: lack of internet link in order to read the PhD thesis.
line 836: lack of internet link to the bibliographic resource.
line 858: link of DOI underlined and in blue (probable text as link)
line 866: link of DOI underlined and in blue (probable text as link)
line 874: link of DOI underlined and in blue (probable text as link)
line 894: Doi without link.
line 902: Doi without link.
line 911: journal name not in italic.
line 917: journal name not in italic.
Pictures:
Figure 2: an identical scale (from 2 to 16) in the Y-axis for both (A) and (B) line charts would help the reader to make a quick visual comparison between the two parts, where in (A) the difference between control and treated is not significant whereas in (B) it is significant. Any reference to 'values not marked by an asterisk' is lacking in the figure caption and should be added.
Suggestion: a colour picture to visually show to the reader the difference between the control and the treated plants in leaf number, growth rate and/or leaf expansion should be added. It would a choice of the authors if in the paper or as Supplementary material.
Author Response
Dear reviewer 05,
Thank you for taking the time to review our manuscript.
The comments and suggestions permitted us to improve and better focalised our manuscript. We carefully considered and provided point-by-point responses to all your comments, corrections and suggestions. Please, see the attachment.
We also provided a new file of the manuscript with all the changes made marked up using the “Track Changes” function in MS Word, so that all modifications can be easily viewed by the reviewers and Editor.
We hope we have satisfied all your questions and suggestions and addressed all the issues raised.
Best regards,
Daniela Di Baccio.
